# An Advanced Study of Urban Emergency Medical Equipment Logistics Distribution for Different Levels of Urgency Demand

**DOI:** 10.3390/ijerph191811264

**Published:** 2022-09-07

**Authors:** Yongqiang Zhao, Liwei Zhang

**Affiliations:** School of Modern Post, Xi’an University of Posts & Telecommunications, Xi’an 710061, China

**Keywords:** major public health emergencies medical supplies, urban emergency logistics, CRITIC method, demand urgency, path optimization

## Abstract

At the early stage of a major public health emergency outbreak, there exists an imbalance between supply and demand in the distribution of emergency supplies. To improve the efficiency of emergency medical service equipment and relieve the treatment pressure of each medical treatment point, one of the most important factors is the emergency medical equipment logistics distribution. Based on the actual data of medical equipment demand during the epidemic and the characteristics of emergencies, this study proposed an evaluation index system for emergency medical equipment demand point urgency, based on the number of patients, the number of available inpatient beds, and other influencing factors as the index. An urban emergency medical equipment distribution model considering the urgency of demand, the distribution time window, and vehicle load was constructed with the constraints. Wuhan, Hubei Province, China, at the beginning of the outbreak was selected as a validation example, and the Criteria Importance Though Intercriteria Correlation (CRITIC) method and the genetic algorithm were used to simulate and validate the model with and without considering the demand urgency. The results show that under the public health emergencies, the distribution path designed to respond to different levels of urgency demand for medical equipment is the most efficient path and reduces the total distribution cost by 5%.

## 1. Introduction

Public health emergencies refer to the sudden occurrence of severe infectious diseases, mass unknown diseases, food and occupational poisoning, and other events that seriously affect public health [1]. At the end of 2019, Coronavirus Disease 2019 (COVID-19) broke out in Wuhan, China, and quickly spread worldwide. During the early stage of the outbreak, a large number of COVID-19 cases emerged in a short period of time, and there was an extreme shortage of medical equipment, protective gear, hospital beds, and healthcare staff. Many infected patients were placed in temporary inpatient areas, which led to severe cross-infection issues. To make the full use of medical resources to ensure that the patients could receive prompt treatment, Wuhan’s government requisitioned a total of 51 hospitals as designated hospitals for COVID-19 patients [2]. However, from January 2020 to March 2020, as medical supplies arrived from all over the country and across the world, the logistics of distributing the supplies were in malfunction. There were many serious problems, including improper allocation of materials and inefficient distribution [3]. This posed a hidden danger to social safety, especially to the patients whose lives depended on the particular medical equipment.

In order to improve the efficiency of emergency relief of urban medical equipment and relieve the treatment pressure of each treatment hospital, medical equipment distribution must simultaneously achieve the goals of rationality and efficiency. Therefore, this study establishes a medical equipment demand urgency evaluation index system to further improve the timeliness and rationality of medical treatment and material transportation, incorporates medical equipment demand urgency as an influencing factor in the distribution path optimization process, and constructs a multi-vehicle emergency medical equipment distribution model that considers demand urgency. A multi-vehicle emergency medical equipment distribution model considering the urgency of demand was constructed, and the priority of hospitals with a higher urgency of equipment demand was improved to ensure the timeliness and rationality of equipment distribution.

## 2. Literature Review

The vehicle routing problem (VRP) of emergency supply distribution has become one of the most popular research topics in recent years. Some studies have focused on improving the timeliness in emergency situations and shortening the distribution time period of emergency supplies [4]. Xu et al. designed an optimization model aiming at the maximum of satisfaction as well as minimum of resource consumption while traffic was under restriction [5]. Research into reduced transportation time has been achieved by different methods, such as testing the rescue path model with multiple objectives with the case of a real urban storm event [6]. Considering uncertain traffic conditions and real road conditions, Sun proposed a bi-objective emergency logistics dispatching model including two objectives, one of which was the transportation time and the other was the transportation cost [7]. The studies mentioned above mainly focused on the timeliness of rescue, ignoring the variability of each demand point and the equity of distribution. On the contrary, more scholars have taken efficiency and fairness as two important elements of disaster relief. Chen et al. proposed a multi-objective optimization model of the distribution of emergency supplies to solve the emergency supplies distribution problem [8]. This model was designed on the basis of the “efficiency, prioritization, and economic dispatch” strategy. Targeting the timeliness and distribution equity of emergency supplies, Qu et al. established a multi-objective planning model for selecting the locations and optimizing the distribution path according to multiple transportation methods and multi-period dynamics [9]. Wu et al. proposed an optimal scheme for dispatching emergency supply vehicles, considering the influence of time windows [10]. The scheme was designed so that the emergency supplies demanded can arrive at their destinations within the required time windows, while lowering the budget as much as possible.

With COVID-19 spreading around the world, the issue of vehicle distribution routes for emergency medical supplies during public health emergencies has aroused more and more scholars’ extensive attention. Unlike normal emergency relief traffic conditions, various traffic barriers did not need to be considered during COVID-19 [11]. Based on that, Hu et al. redesigned the urban emergency medical supplies dispatch and distribution system, constructed a dynamic distribution model for emergency medical supplies, and verified the validity and feasibility of the model by arithmetic examples [12]. Li et al. established a multi-cycle emergency supply distribution model for the panic psychology of patients caused by shortages and the untimely distribution of emergency medical supplies [13]. Zhang et al. constructed a multi-objective emergency medical material dispatching model based on the premise that demand can be split according to the different urgency levels of the material demand points [14]. Chen et al. introduced a polyhedral uncertainty set to portray the uncertainty about the number of casualties, while considering casualty classifications and mobile hospital typology to construct a robust site-selection model with the goal of maximizing the total survival probability of casualties [15]. Some scholars [16,17,18,19,20] also addressed the demand for emergency supplies by studying its classification and the time window constraints for emergency supplies’ demand urgency. Although there has been a foundation for the research related to the emergency logistics of medical supplies during public health emergencies, most of the research has focused on the emergency distribution of consumable medical supplies such as medical protective clothing and medical surgical masks, and there is a shortage of research on the emergency distribution of non-consumable medical equipment such as ventilators and extracorporeal membrane oxygenation (ECMO) devices, and research on the distribution of supplies has mostly been about a single type of distribution vehicle, which is very different from real situations. Therefore, this study constructed a multi-vehicle urban emergency medical equipment distribution model considering the urgency of equipment demand in order to obtain the best medical equipment distribution and transportation plan for major public health emergencies.

The rest of this study is as follows. Section 3 analyzes the urgency of emergency medical equipment demand, Section 4 constructs a model of urban emergency medical equipment distribution considering the urgency of equipment demand, Section 5 conducts an arithmetic analysis using Wuhan City at the beginning of the epidemic outbreak as an example, and Section 6 provides the research conclusions and future research directions.

## 3. Emergency Medical Equipment Demand Urgency Analysis

Wrong decisions may be made and the emergency treatment of patients may be affected if the distribution and path planning of equipment are based only on the demand and distance of the medical equipment demand points. Given the special characteristics of medical equipment distribution during major public health emergencies, it is necessary to prioritize the distribution of equipment demand points with higher demand urgency when equipment reserves are limited and distribution capacity is constrained [21,22].

In this study, the concept of emergency medical equipment demand urgency was proposed and a relatively reasonable emergency medical equipment rescue plan was developed on the basis of actual scenarios of emergency medical equipment demand during the epidemic.

### 3.1. Construction of the Emergency Medical Equipment Urgency Evaluation Index System

Due to the lack of recent research related to medical equipment emergency supply during severe public health emergencies in China, the evaluation index of demand urgency mainly includes the number of affected personnel, the infrastructure, and the material reserves in line with previous experience of general emergency responses [16]. Combined with the experience of the COVID-19 epidemic, there are four factors affecting the urgency of demand for equipment: the number of registered patients, the number of open beds, the number of senior health professionals and technicians, and the demand for medical equipment.
The number of registered patients reflects the basic number of patients. The greater the number of severe patients, the greater the demand for emergency medical equipment. Therefore, the urgency of the demand level for medical equipment is higher.The number of open beds reflects the infrastructure of hospitals. The higher the number of open beds in a hospital is, the larger the hospital is and the more patients it can afford. Therefore, the urgency of the demand level for medical equipment is higher.The number of senior health professionals and technicians reflects the emergency response capacity of each hospital. The more senior health professionals and technicians the hospital has, the more able the hospital is to provide emergency treatment to severely affected patients. Therefore, the urgency of the demand level for medical equipment is higher.The demand for medical equipment reflects the medical supply reserve of each hospital. In this study, the evaluation was mainly based on the gap in medical equipment, i.e., the difference between the level of demand and the level of the inventory. The greater the number of medical equipment gaps a hospital has, the higher the number of medical equipment units needed for that hospital. Therefore, the urgency of medical equipment demand is higher.

### 3.2. Construction of the Emergency Medical Equipment Demand Urgency Model Based on the CRITIC Method

The Criteria Importance Though Intercriteria Correlation (CRITIC) method is a comprehensive objective weight measurement system based on the strength of the comparison and conflict between the evaluation indicators. It considers the influence of variation and the influence of correlation on the indicators. The results are more comprehensive compared with the entropy weight method and the standard deviation method, so for a comprehensive evaluation problem of multiple indicators, using the CRITIC method can eliminate some strong correlations. The basic steps of the CRITIC method to determine the urgency of medical equipment needs in a relief hospital are as follows.

Step 1: Establishing an evaluation index matrix for the urgency of emergency medical equipment needs

Set the set of m rescue hospitals as $X=\left\{x_{1},x_{2},\cdots ,x_{m}\right\}$, where each rescue hospital in the set has *n* evaluation indicators, and let the data of the $j\left(j=1,2,\cdots ,n\right)\mathrm{th}$ impact factor indicator of the *i*th rescue hospital $x_{i}$ be $x_{ij}.$ The data matrix *X* of the emergency medical equipment demand urgency evaluation index is:(1)$$X={\left(x_{ij}\right)}_{m\times n}=\left[\begin{array}{c} x_{11}\\ x_{21}\\ \vdots \\ x_{m1} \end{array}\begin{array}{c} x_{12}\\ x_{22}\\ \vdots \\ x_{m2} \end{array}\begin{array}{c} \cdots \\ \cdots \\ \vdots \\ \cdots \end{array}\begin{array}{c} x_{1n}\\ x_{2n}\\ \vdots \\ x_{mn} \end{array}\right]$$

Step 2: Dimensionless processing

In order to eliminate the influence of the different dimensions on the evaluation results, each evaluation index of the urgency of emergency medical equipment demand needs to be made dimensionless. The general formula is as follows:(2)$$x_{ij}^{*}=\frac{x_{ij}-min\left\{x_{1j},x_{2j},\cdots ,x_{nj}\right\}}{max\left\{x_{1j},x_{2j},\cdots ,x_{nj}\right\}-min\left\{x_{1j},x_{2j},\cdots ,x_{nj}\right\}}$$
where $x_{ij}^{*}$ is the value after dimensionless processing of the data. The matrix *P* after dimensionless processing of the data is:(3)$$P={\left(p_{ij}\right)}_{m\times n}=\left[\begin{array}{cccc} p_{11} & p_{12} & \cdots & p_{1n}\\ p_{21} & p_{22} & \cdots & p_{2n}\\ \vdots & \vdots & \vdots & \vdots \\ p_{m1} & p_{m2} & \cdots & p_{mn} \end{array}\right]$$
(4)$$p_{ij}=x_{ij}^{*},\;i=1,2,3,\cdots m,\;j=1,2,3,\cdots m$$

Step 3: Determine the variability of the evaluation indicators for the urgency of emergency medical equipment needs

The variability of each indicator of the urgency of emergency medical equipment demand should be in the form of a standard deviation, and $S_{j}$ denotes the standard deviation of the *j*th indicator:(5)$$\left\{\begin{array}{c} {\bar{p}}_{j}=\frac{1}{m}\sum_{i=1}^{m}p_{ij}\\ S_{j}=\sqrt{\frac{\sum_{i=1}^{m}{\left(p_{ij}-{\bar{p}}_{j}\right)}^{2}}{m-1}} \end{array}\right.$$

Step 4: Determine the conflict of emergency medical equipment urgency evaluation indexes

The correlation coefficient is a statistical indicator that reflects the closeness of the correlation between variables. The correlation coefficient is calculated by the product–difference method, which is also based on the deviation of two variables from the mean, and reflects the degree of correlation between two variables by multiplying the two deviations. The conflict between the indicators is based on the correlation coefficient between the indicators. For example, a strong positive correlation between the two indicators, illustrating that the conflict between the two indicators is low, reflects that the more similar the information is, the more repetitive the evaluation content will be. To a certain extent, it also weakens the evaluation strength of the indicator, and the weight assigned to the indicator should be reduced. Here, $r_{ij}$ denotes the correlation coefficient between emergency medical equipment demand urgency evaluation indicators *i* and *j*, where $R_{j}$ denotes the quantitative indicator of the conflict between the *j*th emergency medical equipment demand urgency evaluation indicator and other indicators:(6)$$R_{j}=\sum_{i=1}^{m}\left(1-r_{ij}\right)$$

Step 5: Calculate the amount of information for the *j*th emergency medical equipment demand urgency evaluation index

$C_{j}$ is the quantity of information in the *j*th emergency medical equipment demand urgency evaluation index. The greater $C_{j}$ is, the greater the role of the *j*th evaluation index in the overall emergency medical equipment urgency evaluation index system, and the more weight should be assigned to it:(7)$$C_{j}=S_{j}\sum_{i=1}^{m}\left(1-r_{ij}\right)=S_{j}\times R_{j}$$

Step 6: Calculate the weights of each emergency medical equipment demand urgency evaluation index $\alpha_{j}$
(8)$$\alpha_{j}=\frac{C_{j}}{\sum_{j=1}^{n}C_{j}}$$

Step 7: Determine the urgency of the need for the medical equipment demand points

In this study, the general weighted summation method was used to determine the composite score of the urgency of emergency medical equipment demand for each treatment hospital as follows:(9)$$\omega_{i}=\sum_{j=1}^{n}\alpha_{j}p_{ij}$$

Subsequently, according to the emergency medical equipment demand urgency score of each treatment hospital, the treatment hospital with the minimum score was used as the benchmark and compared with the scores of the other treatment hospitals to obtain the relative demand urgency coefficient of each treatment hospital. The specific formula is as follows:(10)$$\lambda_{i}=\frac{\omega_{i}}{\omega_{i\;min}}$$
where $\omega_{i}$ is the demand urgency score of treatment hospital *i*; $\alpha_{j}$ is the weight of each emergency medical equipment urgency evaluation index; $p_{ij}$ is the standardized value of emergency medical equipment urgency evaluation index *j* of treatment hospital *i*; $\lambda_{i}$ is the relative demand urgency of treatment hospital *i*; and $\omega_{i\;min}$ is the minimum demand urgency score of treatment hospital *i*.

## 4. Construction of an Urban Emergency Medical Equipment Distribution Model Considering the Urgency of Equipment Demand

At the early stage of major public health emergencies, the distribution center has limited stores of emergency medical equipment and cannot meet the demand for equipment at all demand points. According to the CRITIC method, we determined the urgency of demand for different equipment at each demand point, and distributed equipment to each demand point according to the urgency levels. Through this method, we ensured the relative fairness of equipment distribution, selected the type of vehicles and distribution paths reasonably, and prioritized distribution to the equipment demand points with higher urgency of demand, thus effectively alleviating the pressure of patient treatment at each demand point. Therefore, this study constructed a multi-model emergency medical equipment distribution model that maximizes the equity of emergency equipment distribution and minimizes the total distribution cost based on the demand urgency.

### 4.1. Model Assumptions

To facilitate the study of the problem, the following assumptions were made for the actual scenario of urban emergency medical equipment distribution during major public health emergencies.
Emergency medical equipment is characterized by the joint participation of government and social resources, and its focus during a public health emergency is the effective utilization of emergency medical resources. Therefore, in order to ensure that the medical equipment demand points with urgent needs obtain sufficient medical equipment as soon as possible, the government or non-profit organizations should develop a corresponding reward and punishment system to improve the priority of distribution to the demand points with urgent needs and reduce distribution delays.The locations of the distribution center and all emergency medical equipment demand points are known.The total amount of equipment stored in the distribution center is less than the total demand at each emergency medical equipment demand point.The demand for equipment at each emergency medical equipment demand point is known.The amount of equipment stored in the distribution center is known.All vehicles depart from the distribution center, deliver all medical equipment to each demand point, and then return to the distribution center.Each emergency medical equipment demand point can only be supplied by one vehicle for distribution.The distribution center has various types of transportation vehicles and sufficient transportation vehicles.The distribution center has limited distribution personnel and should use as few distribution vehicles as possible.Different distribution vehicles have the same activation cost, and the vehicle activation cost is inversely proportional to the number of distribution personnel.The transportation cost per unit of distance of all vehicles is fixed and can be estimated, and the transportation cost per unit of time, the loading capacity, and the driving speed of different models of vehicles are different.Medical equipment loading and unloading time is not accounted for.

### 4.2. Symbol Description

On the basis of the aforementioned assumptions, the parameters and descriptions are shown in Table 1, Table 2 and Table 3.

### 4.3. Model Construction

With the factors mentioned above, the demand urgency of emergency medical equipment demand points was used as an influencing factor to establish a multi-model emergency medical equipment distribution model that prioritizes distribution to demand points with high demand urgency. Due to the urgency of equipment demand, the sooner the equipment arrives, the sooner it is available. Therefore, the soft time window does not set the earliest receiving time; instead, it sets the desired time $LT_{i}$ and the latest delivery time $LLT_{i}$. When the emergency medical equipment arrives before the desired time $LT_{i}$, the government or nonprofit organizations will give additional economic subsidies; when the delivery time is $\left(LT_{i},LLT_{i}\right)$, this is regarded as a distribution delay, which leads to a negative impact for the emergency when the delivery time exceeds $LLT_{i}$. It may cause a serious impact on the emergency medical equipment demand point, and thus a stiff penalty should be applied. The vehicles’ fixed cost, the vehicles’ driving cost, the subsidy incentive cost, the vehicles’ delay cost, and the overtime cost can be obtained as follows:(11)$$C_{1}=Pn$$
(12)$$C_{2}=\sum_{i=0}^{m}\sum_{j=0}^{m}\sum_{u=1}^{f}\sum_{k=1}^{e}\frac{C_{uk}S_{ij}x_{ijuk}}{v_{uk}}$$
(13)$$C_{3}=\alpha \sum_{i=1}^{m}max\left(LT_{i}-t_{i},0\right)\lambda_{i}$$
(14)$$C_{4}=\beta \sum_{i=1}^{m}min\left(t_{i}-LT_{i},LLT_{i}-LT_{i}\right)\theta_{i}$$
(15)$$C_{5}=\gamma \sum_{i=1}^{m}max\left(t_{i}-LLT_{i},\;0\right)$$

An analysis of the component costs shows that achieving the lowest total distribution cost only requires minimizing the sum of the vehicles’ fixed cost $C_{1}$, the driving cost $C_{2}$, the delay cost $C_{4}$, and the overtime cost $C_{5}$, while also maximizing the subsidy cost $C_{3}$ and finally minimizing the sum of the two, so that the objective function and constraints can be constructed as follows.

The objective function is:(16)$$MinZ_{1}=\sum_{i=1}^{m}\omega_{i}\left(q_{i}-h_{i}\right)\;$$
(17)$$\begin{array}{c} MinZ_{2}=Pn+\sum_{i=0}^{m}\sum_{j=0}^{m}\sum_{u=1}^{f}\sum_{k=1}^{e}\frac{C_{uk}S_{ij}x_{ijuk}}{v_{uk}}-\alpha \sum_{i=1}^{m}max\left(LT_{i}-t_{i},0\right)\lambda_{i}\\ +\beta \sum_{i=1}^{m}min\left(t_{i}-LT_{i},LLT_{i}-LT_{i}\right)\theta_{i}+\gamma \sum_{i=1}^{m}max\left(t_{i}-LLT_{i},0\right)\; \end{array}$$

The constraints are:(18)$$\sum_{i=1}^{m}h_{i}y_{iuk}\leq Q_{uk},\;\forall u\in U,\;\forall k\in K\;$$
(19)$$\sum_{i=1}^{m}h_{i}\leq G$$
(20)$$\sum_{i=0}^{m}\sum_{u=1}^{f}\sum_{k=1}^{e}x_{ijuk}=\sum_{i=0}^{m}\sum_{u=1}^{f}\sum_{k=1}^{e}x_{jiuk}=1,\;\forall j\in I$$
(21)$$\sum_{i=1}^{m}x_{0iuk}=\sum_{i=1}^{m}x_{i0uk}=d,\;d\in \left\{0,1\right\},\;\forall u\in U,\;\forall k\in K$$
(22)$$h_{i}\geq 0.6q_{i},\;\left(i=1,2,3,4\cdots \cdots m\right)$$
(23)$$q_{i}\geq h_{i}\geq 0,\;\left(i=1,2,3,4\cdots \cdots m\right)$$
(24)$$t_{j}=\sum_{i=0}^{m}\sum_{u=1}^{f}\sum_{k=1}^{e}x_{ijuk}\left(t_{i}+\frac{S_{ij}}{v_{uk}}\right),\;\forall j\in I,i\neq j$$
(25)$$x_{ijuk}=\left\{\begin{array}{cc} 1 & Type\;u\;vehicle\;k\;travels\;from\;demand\;point\;i\;to\;j\\ 0 & otherwise \end{array}\right.$$
(26)$$y_{iuk}=\left\{\begin{array}{cc} 1 & Demand\;point\;i\;is\;in\;the\;charge\;of\;vehicle\;of\;model\;u\\ 0 & otherwies \end{array}\right.$$
(27)$$\theta_{i}=\left\{\begin{array}{c} 1\;LT_{i}<t_{i}<LLT_{i}\\ 0\;otherwise \end{array}\right.$$

Equation (16) indicates that the urgency and equity of emergency equipment distribution are taken into account so that the distribution plan can achieve equity and maximize utility. Equation (17) indicates that the total distribution cost is minimized. Equation (18) indicates that the actual loading capacity of each distribution vehicle does not exceed its maximum vehicle loading capacity. Equation (19) indicates that the total amount of equipment distributed cannot exceed the total amount of equipment in the distribution center. Equation (20) indicates that the vehicle must leave after delivering materials to the demand point and must not stay (i.e., there must be one entry and one exit for the material demand point). Equation (21) indicates that each vehicle must return to the starting distribution center after completing the distribution task. Equation (22) indicates that 60% of the quantity demanded by the demand point must be met. Equation (23) indicates that the quantity of equipment allocated to each demand point must be less than its demand. Equation (24) indicates the time required to deliver medical equipment to each demand point; and Equations (25)–(27) are the decision variables.

### 4.4. Algorithm Design

In this study, a genetic algorithm was used to solve the emergency medical equipment distribution path model, considering the demand urgency. The flow chart is shown in Figure 1, and the algorithm’s steps are given below.
(1)Formation of the initial population

Firstly, an initial population of size *NP* × *Num* is randomly generated by using real number coding, where *NP* denotes the population size, *Num* denotes the chromosome length, *Num* = *m* + *n*, m is the number of medical equipment demand points, and *n* is the number of vehicles owned by the temporary distribution center. In this study, *NP* = 200, *m* = 12, and *n* = 12. Only one particular chromosome is listed below as an example. As shown in Table 4:

In this chromosome, 1–12 indicate the numbers of the medical equipment demand points, and 13–24 indicate the numbers of the distribution vehicles.

In this scheme, three batches of distribution vehicles are dispatched, and the vehicle paths are 1–4–5–6–11, 10–12–9–3, and 2–8–7 respectively.
(2)Calculate the fitness function

The larger the degree of fitness is, the higher the probability of being selected. The subject of this study was the emergency medical equipment distribution model with capacity constraints, and the penalty cost of violating the vehicle’s own loading capacity constraints was increased when calculating the fitness degree value, and the objective function was to minimize the function value as the model optimization objective. Therefore, the fitness function should be constructed as:(28)$$Fitness_{i}=\frac{1}{C_{i}+PB}$$

In Equation (28), $Fitness_{i}$ denotes the fitness function value of individual gene *i*, $C_{i}$ is the objective function value of individual gene *i*, P=100,000 is the penalty cost of a medical device distribution vehicle violating its own loading capacity, and *B* is a 0–1 variable: when a distribution vehicle violates its own loading capacity in this scheme *B* = 1, and *B* = 0 otherwise.
(3)Selection

The roulette wheel method is used to randomly select the parents.
(4)Crossover

This involves the selection of the two-point crossover, i.e., gene swapping between chromosomal crossovers of the selected parents.
(5)Testing

A test step is performed after the crossover operation to check whether the crossover operation produces valid offspring chromosomes. If there are duplicate genes in the offspring chromosomes, return to Step 4 to reselect the parents.
(6)Mutation

Three variation operators (crossover variation, flip-flop variation, and insertion variation) are used to perform the combined variation operation.

## 5. Example Analysis

### 5.1. Determination of the Demand Urgency of Medical Equipment Demand Points

In this study, the background of emergency medical equipment supply in Wuhan City at the beginning of the COVID-19 epidemic outbreak was applied, and the relevant parameters were combined with actual data and calculations. Wuhan Tianhe International Airport was set as the distribution center, and the 12 designated hospitals that were requisitioned during the COVID-19 epidemic in Wuhan City were set as the demand points for medical equipment. We selected ventilators as the medical equipment for distribution. The original data of the 12 designated hospitals are shown in Table 5: the number of open beds and the number of admitted beds were obtained from the statistics released by the Wuhan Health Care Commission in February 2020. The number of senior health professionals was obtained from the data published by the 12 designated hospitals, and the demand for ventilators was estimated according to the number of newly admitted beds as well as the number of open beds in each sentinel hospital from 3 February to 4 February, with a ratio of 1:1 between new patients and ventilator demand. SPSS statistical analysis software was used to standardize the original data for each medical equipment demand point, and the standardized data obtained are shown in Table 6.

The weight coefficients of the four evaluation indexes were calculated by the CRITIC method as 0.261, 0.274, 0.355, and 0.110 accordingly; subsequently, the general weighted summation method was used to calculate the demand urgency scores of Hospitals 1–12. finally, the hospital with the smallest score was taken as the benchmark and compared with the scores of other hospitals. The results are shown in Table 7. According to the level of relative demand urgency, the ranking of the hospitals in the calculation example was 6, 1, 11, 8, 12, 2, 3, 5, 9, 10, 4, 7.

### 5.2. Urban Emergency Medical Equipment Distribution Model Considering Demand Urgency

#### 5.2.1. Time Window

In urban logistics distribution scenarios, it is generally required that the distribution task must be completed within 2 h. In this study, considering the urgency of demand, the expected time was set as the shortest time required for distribution directly from the distribution center to each medical equipment demand point. The expected time of each demand point was adjusted equally according to the urgency level of equipment demand, followed by cluster analysis of the urgency demand to determine the latest arrival time of each demand point [16].

A cluster analysis of the demand urgency was performed, and the results of the analysis are as follows.

From Figure 2, it can be seen that the optimal cluster number is 2. Therefore, the device demand points were divided into two levels. The first level included demand points Nos 6 and 1, and the second level includes demand points Nos 10, 7, 4, 9, 2, 5, 3, 12, 11, and 8. Since the demand urgency of the equipment was higher, the requirements of the time window were higher, and thus the time interval between $LT_{i}$ and $LLT_{i}$ for the first-level demand points was set to 15 min, and the time interval for the second-level demand points was 30 min. The adjusted time windows of each equipment demand point according to the demand urgency are shown in Table 8 [23].

#### 5.2.2. Parameter Setting

Emergency medical equipment delivery vehicles departed from Wuhan Tianhe International Airport to deliver medical equipment to the 12 designated hospitals. It was assumed that Wuhan Tianhe International Airport had 350 respirators available on that day, and the activation cost of each vehicle was CNY300 uniformly, with three types of delivery vans: a Keizo Isuzu 100P (QL5041XXYBUHAJ), a Keizo Isuzu 700P (QL5100XXYA8LAJ), and a Jiefang flat-head diesel truck CA1125J. The ventilator’s outer packaging box size was 80 × 50 × 50 cm, with a total weight of 17 kg. The maximum loading capacity of each type of vehicle was calculated, the average speed of the vehicle was taken as 50% of the maximum speed, and the vehicle travels one unit of distance, with the cost calculated with reference to the fuel consumption of the vehicle per 100 km. The specific parameters of the vehicle are shown in Table 9, the other parameters are displayed in Table 10. The shortest distance between the distribution center and the hospital is shown in Table 11.

#### 5.2.3. Analysis of the Calculated Results

The CRITIC method and the genetic algorithm were implemented programmatically using MATLAB 2018a. The hardware environment used for the experiments was an Intel(R) Core (TM) i7-9700 CPU @ 3.00 GHz, 16 GB RAM, and a 64-bit Windows 10 operating system.

We obtained optimal distribution paths programmed by MATLAB 2018a: Type 2 vehicles follow Route 1, which pass the demand points marked 9, 11, 4, 10, 5, and 8 accordingly, and return to the distribution center after the distribution has been completed. Another path is that Type 2 vehicles follow Route 2, passing the demand points marked 3, 6, and 7 accordingly, and returning to the distribution center after the distribution has been completed. The third path is that Type 3 vehicles follow Route 3, which pass the demand points marked as 1, 2, and 12 accordingly, and return to the distribution center after the distribution has been completed. For the distribution order, the demand points with higher demand urgency received priority distribution, and the total distribution cost was CNY1108.258, as shown in Table 12.

To further verify the validity of the model, MATLAB was also used to calculate the emergency medical equipment vehicle path model without the variable of demand urgency. Therefore, we set the demand urgency of all emergency equipment demand points to 1, resulting in the following distribution paths: Type 2 vehicles follow Route 1, passing through the demand points marked 9, 11, 4, 5, 8 and 10 accordingly, and then returning to the distribution center after the distribution has been completed; Type 2 vehicles follow Route 2 and pass through the demand points marked 3, 7, and 6, and then return to the distribution center; Type 2 vehicles follow Route 3 and pass through the demand points marked 2, 1, and 12, and then return to the distribution center. There was no priority given to the equipment demand points with higher demand urgency and the total distribution cost was CNY1504.649. Since the government subsidy does not count when demand urgency is not prioritized, based on the determination of the distribution route, the government subsidy was calculated separately and deducted from the total distribution cost, which led to the final distribution cost: CNY1167.07. This is still greater than the total distribution cost when we take the equipment demand urgency into consideration, as shown in Table 13.

By comparing Table 12 and Table 13, it was found that considering the demand urgency reduced the total distribution cost by 5% and increased the total distribution time by 2.3% compared with the emergency medical equipment distribution model without considering the demand urgency. This improved the distribution priority of the hospitals with high demand urgency in terms of the distribution order. Unlike the general path optimization model, the model constructed in this study assumes that the government has developed an incentive mechanism based on positive encouragement in order to incentivize logistics companies to minimize distribution delays and to consider the differences in the demand urgency of the treatment hospitals in the distribution process. Under this assumption, the cost function consists of the driving cost and the reward and penalty cost, and the objective function is to minimize the total cost. This was designed to achieve the goal of considering the timeliness of delivery and taking the reasonableness of the delivery order into account, and to give priority to the hospitals with high demand urgency as much as possible to ensure the emergency treatment of patients. Therefore, the model constructed in this study can increase the delivery time to some extent compared with the emergency medical equipment delivery model without considering the urgency of demand, thus obtaining greater benefits. 

#### 5.2.4. Discussion

This study presented an emergency medical equipment distribution path problem with a single distribution center and multiple types of vehicles, and the demand urgency was brought into the construction of the model as an influencing factor, which took the distribution’s rationality into account while considering the distribution’s timeliness, and provides a reference for solving the urban emergency medical equipment distribution problem. Compared with previous studies [24], the model constructed in this study is more in line with the actual situation. In terms of applying the method, the CRITIC method and realistic data were used to calculate the urgency of emergency medical equipment demand, which is more convincing than the results obtained by AHP and hypothetical data in previous studies [24]. The model constructed in this study is applicable to the case of multiple models with a single distribution center and indivisible demand. In reality, there are more categories and quantities of emergency medical equipment in the distribution of emergency medical equipment, and the subsequent study of the distribution of emergency medical equipment in the case of diverse types, multiple distribution centers, and splittable demand should be conducted to improve the general applicability of the model.

## 6. Conclusions

(1)In this study, the concept of equipment demand urgency is introduced against the background of the urban logistics distribution problem of public health emergencies. We analyzed the factors related to equipment demand urgency of medical equipment demand points, and designed an evaluation index system for the demand urgency of emergency medical equipment demand points. This study calculated the weight of each relevant index by the CRITIC method, and the weight of the urgency of equipment demand of each medical aid point, compared with the urgency of equipment demand obtained by AHP and other subjective weighting methods, were more rigorous and accurate, and avoided the interference of subjective factors of relevant personnel.(2)In terms of solving the model, this study solved the optimal distribution path model considering and not considering the equipment demand urgency. The calculation results show that the emergency medical equipment distribution model considering the demand urgency not only prioritizes the distribution of medical equipment with higher demand urgency, but also optimizes the distribution time and minimizes the total distribution cost, which verifies the validity of the model and proves the value of the proposed model.(3)Based on previous research, this study constructed an emergency medical equipment urban distribution model considering multiple vehicle models with medical equipment as the transportation cargo, which not only makes the selection of the distribution vehicles more flexible, but also improves vehicle utilization and the distribution efficiency, and has greater application value. In reality, when emergency medical vehicles are distributed, there are situations where the demand for equipment can be split, many kinds of equipment can be transported, and multiple distribution centers operate simultaneously. In order to be more in line with the actual situation, the subsequent in-depth study of the emergency medical supply distribution problem in the situation of different emergency medical supplies, multiple supply distribution centers, and demand that can be split should be conducted to enhance the practicality of the model.

## Figures and Tables

**Figure 1 ijerph-19-11264-f001:**
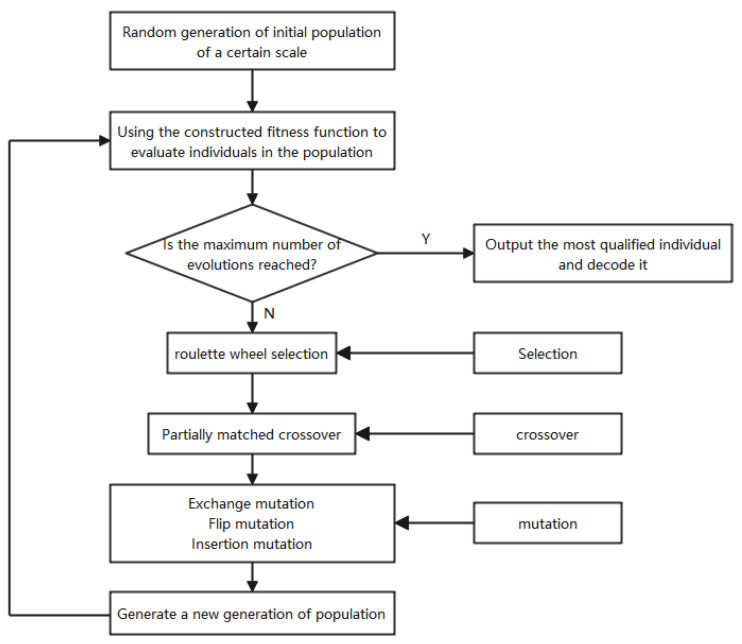
Operation process of the genetic algorithm.

**Figure 2 ijerph-19-11264-f002:**
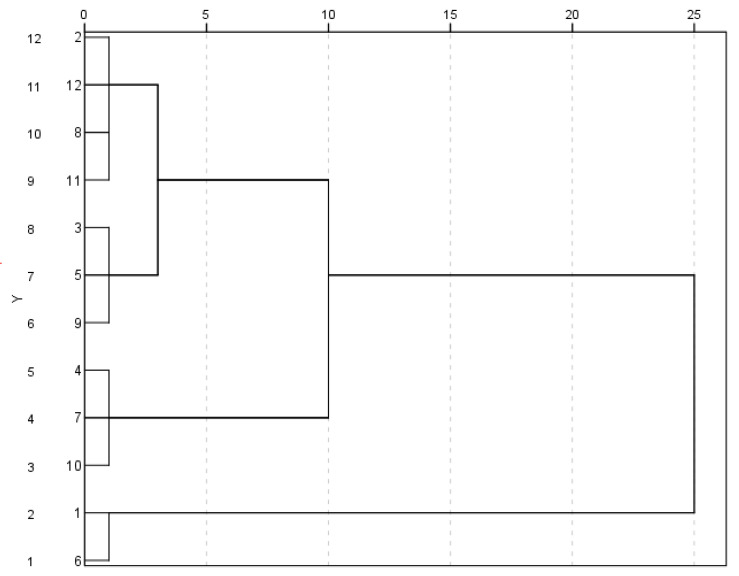
Cluster analysis results graph.

**Table 1 ijerph-19-11264-t001:** Set symbols and descriptions.

Set	Descriptions
I	$\mathrm{Set}\;\mathrm{of}\;\mathrm{network}\;\mathrm{node}\;\mathrm{numbers},\;\left(i\in I,I=0,1,2,3,4\cdots \cdots m\right),\;\mathrm{where}\;0\;\mathrm{is}\;\mathrm{the}\;\mathrm{distribution}\;\mathrm{center}\;\mathrm{and}\;1,2,3,4\cdots \cdots m\;\mathrm{are}\;\mathrm{the}\;\mathrm{emergency}\;\mathrm{medical}\;\mathrm{equipment}\;\mathrm{demand}\;\mathrm{points}$
K	$\mathrm{Vehicle}\;\mathrm{number}\;\mathrm{set},\;\left(k\in K,K=1,2,3,4\cdots \cdots e\right)$
U	$\mathrm{Vehicle}\;\mathrm{type}\;\mathrm{set},\;\left(u\in U,U=1,2\cdots \cdots f\right)$

**Table 2 ijerph-19-11264-t002:** Parameters’ symbols and descriptions.

Parameters	Descriptions
$C_{uk}$	Cost per unit of distance traveled for vehicle *k* of type *u*
$S_{ij}$	The shortest driving distance from demand point *i* to *j*
α	Government subsidy factor
β	Penalty coefficient for distribution delays per unit of time
γ	Penalty factor for distribution overruns per unit of time
G	The amount of emergency medical equipment available at the distribution center
$q_{i}$	The level of demand for emergency medical equipment at demand point *i*
$h_{i}$	The amount of emergency medical equipment distributed at demand point *i*
$Q_{uk}$	The maximum load capacity of vehicle *k* of type *u*
$v_{uk}$	The average travel speed of vehicle *k* of type *u*
$t_{i}$	The actual arrival time of the distribution vehicle at demand point *i*
P	The activation cost of the vehicle
n	The number of times the vehicle is activated

**Table 3 ijerph-19-11264-t003:** Variables’ symbols and descriptions.

Variables	Descriptions
$x_{ijuk}$	$0–1\;\mathrm{variable},\;x_{ijuk}$ is 1 when vehicle *k* of type *u* travels from demand point *i* to *j*, and 0 otherwise
$y_{iuk}$	$0–1\;\mathrm{variable},\;y_{iuk}$ is 1 when the demand point *i* is the responsibility of vehicle *k* of type *u*, and 0 otherwise
$\theta_{i}$	$0–1\;\mathrm{variable},\;\mathrm{when}\;t_{i}<LT_{i}$$\theta_{i}$ is 0, and 1 otherwise

**Table 4 ijerph-19-11264-t004:** Example.

14	15	1	4	5	6	11	13	10	12	9	3	16	2	8	7	17	18		21	19	23	20	24	22

**Table 5 ijerph-19-11264-t005:** Information of each medical equipment demand point.

No	Designated Hospital	Number of Open Beds	Number of Admitted Beds	Number of Middle and Senior Health Professionals	Ventilator Demand
1	Wuhan Jinyintan Hospital	720	688	169	62
2	Wuhan Hankou Hospital	389	435	265	46
3	Wuhan Fifth Hospital	430	427	228	6
4	Wuhan Seventh Hospital	220	233	220	37
5	Wuhan Third Hospital (Guanggu Campus)	300	318	357	6
6	Tongji Hospital Sino-French New City Campus	360	340	582	89
7	Wuhan Union Medical College Hospital (West Hospital)	267	267	175	20
8	Hubei Provincial People’s Hospital (East Hospital)	288	288	508	18
9	Hubei Hospital of Integrative Medicine	186	186	460	13
10	Tianyou Hospital, Wuhan University of Science and Technology	265	265	240	33
11	Wuhan Sixth Hospital	390	406	370	22
12	Wuhan Huangpi District Traditional Chinese Medicine Hospital	442	442	262	26

Note: Data were obtained from the official website of Wuhan Municipal Health and Wellness Commission.

**Table 6 ijerph-19-11264-t006:** Standardized information for each medical equipment demand point.

No	Designated Hospital	Number of Open Beds	Number of Admitted Beds	Number of Middle and Senior Health Professionals	Ventilator Demand
1	Wuhan Jinyintan Hospital	1.000	1.000	0.000	0.675
2	Wuhan Hankou Hospital	0.380	0.496	0.232	0.482
3	Wuhan Fifth Hospital	0.457	0.480	0.143	0.000
4	Wuhan Seventh Hospital	0.064	0.094	0.123	0.373
5	Wuhan Third Hospital (Guanggu Campus)	0.213	0.263	0.455	0.000
6	Tongji Hospital Sino-French New City Campus	0.326	0.307	1.000	1.000
7	Wuhan Union Medical College Hospital (West Hospital)	0.152	0.161	0.015	0.169
8	Hubei Provincial People’s Hospital (East Hospital)	0.191	0.203	0.821	0.145
9	Hubei Hospital of Integrative Medicine	0.000	0.000	0.705	0.084
10	Tianyou Hospital, Wuhan University of Science and Technology	0.148	0.157	0.172	0.325
11	Wuhan Sixth Hospital	0.382	0.438	0.487	0.193
12	Wuhan Huangpi District Traditional Chinese Medicine Hospital	0.479	0.510	0.225	0.241

**Table 7 ijerph-19-11264-t007:** Demand urgency scores for each medical device demand point.

NO	1	2	3	4	5	6	7	8	9	10	11	12
**Demand urgency score**	0.609	0.371	0.301	0.127	0.289	0.635	0.107	0.413	0.26	0.179	0.414	0.371
**Relative demand urgency**	5.665	3.449	2.803	1.185	2.693	5.905	1	3.843	2.417	1.662	3.85	3.453

**Table 8 ijerph-19-11264-t008:** Time window information for each medical device demand point.

NO	1	2	3	4	5	6	7	8	9	10	11	12
**Expected time (h)**	0.73	0.86	1.08	1.43	1.42	0.98	1.50	1.38	0.99	1.38	0.81	0.86
**Latest time** **(h)**	0.98	1.36	1.58	1.93	1.92	1.23	2.00	1.88	1.49	1.88	1.31	1.36

**Table 9 ijerph-19-11264-t009:** Vehicle parameters.

	Qingling Isuzu 100P (QL5041XXYBUHAJ)	Qingling Isuzu 700P (QL5100XXYA8LAJ)	Jiefang Flathead Diesel Truck CA1125J
**Vehicle type**	1	2	3
**Cargo hold size**	4.25 × 2.05 × 2.1 (m)	5.653 × 2.278 × 2.04 (m)	6.2 × 2.3 × 2.6 (m)
**Rated weight**	1495 kg	3255 kg	6000 kg
**Maximum speed**	105 (km/h)	110 (km/h)	100 (km/h)
**Fuel consumption per 100 km**	14 L	20 L	34 L
**Vehicle start-up cost**	300	300	300
**Average city driving speed**	52.5 (km/h)	55 (km/h)	50 (km/h)
**Vehicle travel cost per unit of distance**	1.12	1.6	2.71
**Maximum loading capacity**	80	112	144

Note: Data source: trucking home.

**Table 10 ijerph-19-11264-t010:** Other parameters.

Parameter	Value
Overrun penalty cost per unit of time (γ)	1000
Delay penalty cost per unit of time (β)	25
Subsidy incentive cost per unit of time (α)	35

**Table 11 ijerph-19-11264-t011:** The shortest distance between the distribution center and the sentinel hospitals.

Demand Point	0	1	2	3	4	5	6	7	8	9	10	11	12
0	0	24	26	32	38	50	39	40	57	24	39	26	26
1	24	0	7.7	18	22	33	26	27	41	9.2	24	9.7	31
2	26	7.7	0	12	14	24	25	23	31	4.4	15	4.3	34
3	32	18	12	0	9.2	17	17	14	28	8.3	9.3	8.4	44
4	38	22	14	9.2	0	10	24	21	21	12	2.8	10	46
5	50	33	24	17	10	0	34	30	8.6	24	11	20	54
6	39	26	25	17	24	34	0	16	41	21	25	23	55
7	40	27	23	14	21	30	16	0	34	20	24	21	53
8	57	41	31	28	21	8.6	41	34	0	34	20	30	62
9	24	9.2	4.4	8.3	12	24	21	20	34	0	13	2.6	37
10	39	24	15	9.3	2.8	11	25	24	20	13	0	12	48
11	26	9.7	4.3	8.4	10	20	23	21	30	2.6	12	0	36
12	26	31	34	44	46	54	55	53	62	37	48	36	0

Note: Data source: Gaode Map.

**Table 12 ijerph-19-11264-t012:** Calculated results considering the demand urgency.

Line Number	Vehicle Type	Number of Equipment Loads (Unit)	Distance Traveled (km)	Travel Path	Activation Cost (CNY)	Driving Cost (CNY)	Government Subsidy Penalty Cost (CNY)	Total Program Distribution Cost (CNY)
1	2	109	116	0→9→11→4→10→5→8→0	300	185.6	249.984	1108.258
2	2	107	105	0→3→6→7→0	300	168	78.365	
3	3	134	91.7	0→1→2→12→0	300	248.507	65.5	

**Table 13 ijerph-19-11264-t013:** Calculation results without considering demand urgency.

Line Number	Vehicle Type	Number of Equipment Loads (Unit)	Distance Traveled (km)	Travel Path	Activation Cost (CNY)	Driving Cost (CNY)	Government Subsidy Penalty Cost (CNY)	Total Program Distribution Cost (CNY)
1	2	109	114.2	0→9→11→4→5→8→10→0	300	182.72	227.861	1167.07
2	2	107	101	0→3→7→6→0	300	161.6	68.421	
3	3	134	90.7	0→2→1→12→0	300	245.797	41.297	

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
