# Peer review of "An Advanced Study of Urban Emergency Medical Equipment Logistics Distribution for Different Levels of Urgency Demand"

_ijerph, 2022, doi:10.3390/ijerph191811264_

Round 1
Reviewer 1 Report
This study proposes an evaluation index model for emergency management. The manuscript is well-written and covers an interesting topic. Here are my comments:
1. What is “arithmetic examples” as addressed in the abstract. These terms are vague and need further explanations (Line 19)
2. What does “allocation of equipment” means in introduction (line 45). I mean allocation to what? Cities, patients, districts, hospitals and so on. This information should be provided in the introduction.
3. Do we really need the last sentence of the introduction (with brackets)? Introduction would be more sound if the followed research design is also addressed.
4. Please define all the abbreviations in their first use (such as VRP – Line 52)
5. From four factors “Lines 128-144” I understand that the distribution problem is for hospitals. Please state this explicitly before this section.
6. What is the definition of CRITIC, please provide its full name.
7. Why CRITIC is superior to the Entropy? Who found that? (Line 147). Please provide more explicit reasons and reference.
8. Section 3.2 only explains CRITIC. Please explain how did you implement CRITIC method in your problem.
9. Please state somewhere in the manuscript about the software you used for CRITIC and genetic programming.
10. Multi-criteria decision making methods (such as AHP or TOPSIS) might have been used to solve this problem as well. Why did not you consider them. What are the advantages of your methodology compared to AHP and TOPSIS?
11. Despite results are satisfactory, I could not see any discussion in the manuscript. Please discuss your findings. Please find similar research and compare your results and methodology with them. Please justify your findings. Please clearly address how your research can be used in practice. A new section is required in this vein.
Reviewer 2 Report
Well structured and well presented model, only need some minor revisions.
Lines 98 – 101: It is better to disengage literature review from the introduction to the policy approach
Lines 141 – 144: Needs further explanation (demand is relating to inventory level and inventory policy)
Lines 169 – 175: To be rephrased to clearly express the role of the correlation coefficient
Lines 340 – 345: Explain more explicitly the combination of the “Demand Urgency Score” and the “Relative Urgency Demand” values
Lines 352 – 355: To be explained better (“The expected time…demand point”)
Lines 363 – 367: “Since…Table 7” to be more clearly explained
Lines 420 -424: “In fact…model” to be further elaborated and substantiated
Lines 441 – 445: Point #3 of conclusions needs to be better substantiated.
